



# Retrieval of thermodynamic profiles in the lower troposphere from GNSS radio occultation using deep learning

Matthias Aichinger-Rosenberger[1] and Jeremiah Sjoberg[2]

[1]Chair of Space Geodesy, Institute of Geodesy and Photogrammetry, ETH Zürich, 8093 Zurich, Switzerland
[2]Constellation Observing System for Meteorology Ionosphere and Climate (COSMIC) Program , University Corporation for Atmospheric Research (UCAR), Boulder, Colorado, USA

**Correspondence:** Matthias Aichinger-Rosenberger, maichinger@ethz.ch

**Abstract.** Global Navigation Satellite Systems (GNSS) radio occultation (RO) is one of the most vital remote sensing techniques globally and of major importance for numerical weather prediction (NWP) and climate science. However, retrieving profiles of atmospheric quantities such as temperature or humidity from GNSS observations is not straightforward and dedicated algorithms still have their limitations. One of these limitations is the need for external meteorological data in the retrieval
process. Various new RO missions have led to an enormous increase in data amounts and with over 10000 globally-distributed, daily profiles, RO can be considered big data nowadays. In this study, we make use of this fact by developing a new retrieval method based on a deep learning model, which only needs RO-specific quantities as an input to produce atmospheric profiles. The model is trained on almost a full year of data from COSMIC-2 and Spire RO missions, using vertical profiles of bending angle (BA) and other RO parameters as input features and operational results from a standard retrieval algorithm as target
values for supervised learning. Initial results from both internal and external validation using reanalysis and radiosonde data suggest that this method produces results with an accuracy comparable to standard algorithms, while mitigating the need for external information in the retrieval process itself. These initial results serve as a starting point for further development of data-driven models for RO, which could significantly enhance the quality of RO products utilized in, e.g., climate sciences by mitigating external biases and increasing independence from other techniques.

## 1 Introduction

Global Navigation Satellite Systems (GNSS) radio occultation (GNSS-RO, hereafter referred to as RO) is one of the most promising remote sensing techniques for global atmospheric sounding (Kursinski et al., 1997; Anthes, 2011; Bonafoni et al., 2019). RO is a limb-sounding technique that uses GNSS signals, which are refracted during their propagation through the Earth's atmosphere to a receiver on a Low-Earth Orbit (LEO) satellite. An occultation event occurs when a GNSS satellite
sets behind (or rises from behind) the horizon and the atmosphere is scanned vertically through the relative movements of the satellites (Steiner et al., 2020). Over the last decades, RO has been extensively used for data assimilation (DA) in numerical weather prediction (NWP) (e.g. Ruston et al. (2022)), where numerous studies have shown large beneficial impacts on global and regional forecasts of high-impact weather phenomena such as tropical cyclones (Huang et al., 2005; Miller et al., 2023; Teng et al., 2023) or atmospheric rivers (Ma et al., 2011). Furthermore, thermodynamic profiles from RO are a vital data source





for climate sciences (Ho et al., 2009; Steiner et al., 2011; Ao et al., 2012; Gleisner et al., 2022)) and probing of the ionosphere (Schreiner et al., 1999; Syndergaard et al., 2006; Lei et al., 2007).

Since the start of the pioneering GPS-MET RO mission in 1996 (Ware et al., 1996), several follow-up missions such as FORMOSAT-3/Constellation Observing System for Meteorology, Ionosphere and Climate (COSMIC; Anthes et al. (2008)) and its successor COSMIC-2 (C2; Schreiner et al. (2020); Weiss et al. (2022)) have been initiated. Additionally, several commercial

companies such as Spire Global have launched their own RO payloads onboard cube satellites. This led to an enormous increase in data amounts, and with over 10000 globally-distributed profiles daily, RO can be considered big data nowadays. Furthermore, the International Radio Occultation Working Group (IROWG) community has recently initiated a collaborative effort to explore the impact of RO observations on NWP: the Radio Occultation Modeling EXperiment (ROMEX, Anthes et al. (2024)). It seeks to quantify the benefit of increasing the number of RO observations using additional observations that are not

available to weather centers for real-time operational systems, such as observations from the commercial missions mentioned above.

These large data amounts make the technique suitable for the application of data-driven models. However, while methods of artificial intelligence (AI) such as machine learning (ML) models, including deep learning (DL), are widely used within various scientific research fields nowadays, they have not been utilized much for RO. Most of the studies making use of AI for RO

research focus on the application of such models to RO products, e.g., for air-quality forecasts (Li et al., 2022), wind field estimation (Chu et al., 2022), 2D mapping of RO observations (Shehaj et al., 2025) as well as the detection of volcanic clouds (Hammouti et al., 2024) or ionospheric scintillation (Ji et al., 2023). Besides this, AI techniques have not been used much in the retrieval process yet, with only few studies which have been conducted in this direction. Although some promising results have been achieved by these studies, they were not able to uncover the full potential of ML, mostly due to the small amounts of data

used. Early studies such as Bonafoni et al. (2009) or Pelliccia et al. (2011) used fewer than 5000 total training profiles, which is less than the current daily number of profiles available from C2 alone. Shyam et al. (2013) used COSMIC observations in an artificial neural network (ANN) framework to retrieve water vapor pressure over the Indian subcontinent with a vertically averaged root mean squared error (RMSE) of 0.57 hPa. They also tested differences in performance between different feature setups using either refractivity or bending angle as model input. Results showed RMSE differences between refractivity and

bending angle inputs below 1.5 and 5 hPa, respectively, with respect to operational products. A fairly recent work by Lasota (2021) utilized 15 months of C2 data and assessed the retrieval performance of an ANN and a random forest (RF) regressor in the lower troposphere by using vertical profiles interpolated from European Centre for Medium-Range Weather Forecasts (ECMWF) Reanalysis v5 (ERA5) (Hersbach et al., 2020) as target values. This approach achieved good accuracy both in the internal validation and in comparison to radiosonde observations (RAOB). Validation using collocated RAOB soundings and

one-dimensional variational assimilation (1D-Var) RO products showed slightly larger discrepancies for the ML models with mean RMSE of around 1.9 K, 1.9 hPa, and 0.5 hPa for the temperature, pressure, and water vapor, respectively. Most recently, Hooda et al. (2023) showed further improvements in the retrieval of water vapor pressure in the lower troposphere over the Indian region by using an ANN trained using C2 observations in a similar approach as Lasota (2021).

All of these previous studies are restricted to particular seasons, geographical regions, parameters or limited amounts of data,



or both. In this study, we propose a new retrieval method using a DL framework for the direct retrieval of temperature, pressure, and specific humidity profiles in the neutral atmosphere, hereafter referred to as AROMA (Advancing the GNSS-RO retrieval of atmospheric profiles using MAchine-learning). As in some of the previous studies, we employ an ANN which has been trained on approximately 1.5 million RO profiles from the C2 and Spire Global RO missions from the year 2022. However, in contrast to other studies, we utilize 1D-Var products for pressure, temperature, and specific humidity as targets, i.e., ground

truth, for model training. We investigate whether this method can serve as an alternative to the 1D-Var technique, which is used by most RO data processing centers today. The main advantage is the retrieval's independence of external meteorological data and of background and observation error covariances while still guaranteeing a similar level of performance.

## 2  Data

This study uses a number of different data sets, utilized for model training, testing, and external validation of results, which are

introduced in the following sections.

### 2.1  RO profiles

The C2 constellation consists of 6 satellites in low-inclination orbits, which produce approximately 5000 atmospheric soundings a day between 45°S and 45°N. It represents a considerable advancement from former missions, especially because of the significantly higher signal-to-noise ratio (SNR) of observations, which are collected by a high-gain, beamforming RO antenna.

This reduces systematic errors such as thermal noise and allows for larger penetration depths in the lower troposphere, with over 50% of profiles reaching below 200 m above the Earth's surface (Lasota, 2021). Various studies showed the high quality of C2 data, confirming its status as state-of-the-art RO data nowadays (Schreiner et al., 2020; Ho et al., 2020; Weiss et al., 2022).

Spire Global, Inc., produces occultations from multiple CubeSats and provided approximately 8000–10000 profiles per day

in January 2021 (Sjoberg et al., 2023). Spire data have been analyzed in various studies over the last years (Bowler, 2020a; Ho et al., 2023; Qiu et al., 2023). These studies confirmed that Spire RO profiles are of generally high quality and confirmed benefits of their usage in NWP models. Although validation results are comparable to those of C2 observations, Spire data have shown somewhat larger uncertainties than other RO missions above 35 km, due to their smaller satellites and antennas, lower SNR, and larger clock errors (Sjoberg et al., 2023). Between 10 and 35 km, slightly smaller uncertainties have been found, a

fact which might be related to larger vertical smoothing (Bowler, 2020b).

In order to obtain global coverage with relatively uniform horizontal resolution, C2 and Spire data are combined into one dataset in this study. This is based on the fact that the latitudinal distributions of the C2 and Spire profiles are quite different (see Figure 1a in Sjoberg et al. (2023)). Spire satellites primarily operate in polar orbits (≥52° inclination) providing global coverage, while C2 satellites are in 24° inclined orbits.

For validation purposes, we additionally use data collected by the constellation of PlanetiQ receivers. From 2020 to 2023, PlanetiQ has launched four RO satellites, GNOMES-1/2/3/4 (GNSS Navigation and Occultation Measurement Satellites), which



employ a Pyxis receiver as a payload. These receivers are capable of tracking GPS, GLONASS, Galileo, and BeiDou GNSS signals, which allows each receiver to obtain about 2500 RO profiles daily (Mo et al., 2024). Although PlanetiQ data has been evaluated by very few studies so far, these show promising results in comparisons to ERA5 and RAOB data (Mo et al., 2024; 95   Ahmed et al., 2024).

In this study we utilize level-2 products from those three missions, in particular data provided in the atmPrf and wetPf2 file formats. The atmPrf file format contains vertical profiles of bending angle (BA) and local spectral width (LSW, see Section 3.4 for details) as well as dry refractivity, geographical coordinates, and various additional data. The wetPf2 files include vertical profiles of temperature, pressure, and water vapor with 100-m vertical sampling derived by the 1D-Var ap-
proach with short-term forecast fields from the European Center for Medium-Range Weather Forecast (ECMWF) operational model used as background. All of these RO profiles are processed and made available in near-real time by the COSMIC Data Analysis and Archive Center (CDAAC) and specific information about the file formats can be found on their homepage (https://www.cosmic.ucar.edu/what-we-do/data-processing-center/data). Since different versions of Spire data are available, we only utilize results computed using the standard UCAR CDAAC processing, also used for processing C2 data, in order to
ensure consistency. It has been shown that this greatly reduces differences in uncertainties and penetration depths, possibly resulting from different processing strategies (Weiss et al., 2022).

## 2.2   ERA5 reanalysis

The ERA5 data set (Hersbach et al., 2020) is the fifth-generation ECMWF atmospheric reanalysis. It provides hourly estimates for a large number of atmospheric, land, and oceanic climate variables with 31-km spatial resolution. The data covers the period
from 1940 to the present and can be accessed through the Climate Data Store (Copernicus Climate Change Service, 2018).

For this study, ERA5 hourly pressure, temperature, and specific humidity data from 30 days (05.04. - 05.05.2023) have been used for external validation of the AROMA retrieval performance. From these hourly 3D data, vertical profiles at the RO profile locations are derived using the interpolation strategies described in Section 3.6.

## 2.3   Radiosondes

In addition to ERA5, a second external validation of AROMA results is carried out using a dataset of global radiosonde observations (RAOBs) collected by ECMWF. We take only those RAOB profiles that have finer than 100-m sampling and reach at least 30 km altitude.

The collocation criteria employed in this study are the following: (1) a maximum horizontal distance of 500 km and (2) a maximum time difference of three hours between the respective RO profile and RAOB sounding.





## 3 Methodology

In this section, we provide a brief introduction to both the classic retrieval of temperature, pressure, and humidity profiles from RO observations as well as the DL approach employed in this study, including the utilized setup of input features for the AROMA model, the specific values chosen for its hyperparameters, and pre-processing strategies.

### 3.1 Standard moist air RO retrieval

The retrieval of atmospheric profiles from GNSS observations is based on accurately measuring phase deviations (called excess phase), which are induced by atmospheric bending of the signal. These excess phase and derived Doppler shift observations of the refracted signals rely on precise atomic clocks utilized by GNSS. This enables traceability to the SI unit of time and thus, RO observations are unbiased, stable across multiple RO missions, and highly accurate. The Doppler shift allows for the computation of atmospheric BA, which depends on atmospheric refractivity (N). By assuming a spherically symmetric atmosphere, a high-resolution vertical refractivity profile can then be derived by applying the inverse Abel transform (Fjeldbo et al., 1971) on the bending angle profile:

$$n(r) = \exp\left[\frac{1}{\pi}\int\limits_{x}^{\infty}\frac{\alpha(a)}{\sqrt{a^2 - x^2}}da\right] \tag{1}$$

where the refractive index equals $n = 1 + N \cdot 10^{-6}$ and $x = r \cdot n(r)$ is a refractional radius. This refractivity profile is related to temperature (T), pressure (p), and water vapor pressure (e) via:

$$N = k_1\frac{p}{T} + k_2\frac{e}{T^2} \tag{2}$$

where $k_1$ = 77.6 K hPa$^{-1}$, $k_2$ = 3.73 x $10^5$ K$^2$hPa$^{-1}$. The units of T, p, and e are K, hPa, and hPa, respectively.

Vertical profiles of upper atmospheric temperature and pressure can then be retrieved from N using the first term of Equation 2 (i.e., assuming no moisture), the hydrostatic integral, and ideal gas law (Kursinski et al., 1997).

In the lower troposphere however, the effects of moisture must be considered. This leads to a more complicated retrieval process, which needs a-priori information about temperature, pressure, and/or water vapor pressure to separate their contributions to the refractivity equation. Various solutions have been proposed in literature, which we briefly summarize. For more extensive overviews, the reader is referred to Lasota (2021) or Li et al. (2019).

The earliest wet retrieval methods were direct approaches, which exploit external pressure and temperature profiles from radiosondes observations or weather models (Kursinski et al., 1997; Ware et al., 1996; Kursinski et al., 1995). They have the drawback of relying on the assumption that these background data are 'error-free', resulting in unrealistic uncertainty estimates. These effects are reduced by using one-dimensional variational (1D-Var) retrieval methods, where RO measurements are combined with background information from NWP in a statistically optimal way (Healy and Eyre, 2000; Poli et al., 2002; Wee et al., 2022). The 1D-Var algorithm is based on the minimization of the cost function

$$J(x) = \frac{1}{2}(x - x_B)^T B^{-1}(x - x_B) + \frac{1}{2}(y_o - H[x])^T O^{-1}(y_o - H[x]) \tag{3}$$



where $H[x]$ denotes a forward operator mapping the state $x$ to the observation space $y_o$. The matrices $B$ and $O$ are background and observation error covariance matrices, respectively, representing the standard uncertainties and correlations of the background data and the observation (plus forward-modeled) data. By variation of the state vector $x$, $J(x)$ is minimized and thus ensures that the final retrieved state $x_r$ minimizes the total deviation against both background and observational data.

This method is the most commonly used one at RO processing centers, such as CDAAC (Wee et al., 2022), the Radio Oc-
cultation Meteorology Satellite Application Facilities (ROM-SAF) (ROM SAF, 2018) , and Jet Propulsion Laboratory (JPL) (Vergados et al., 2014). The accuracy and reliability of these retrievals has been analyzed by various studies, using simulations, NWP models and reanalyses, or radiosonde data (Steiner and Kirchengast, 2005; Ho et al., 2010). For further details, the reader is referred to a number of works in literature detailing the full retrieval process (Kursinski et al., 1997; Steiner et al., 1999; Kuo et al., 2004)), as well as 1D-Var implementations at specific processing centers (Wee et al., 2022; Vergados et al., 2014;
Li et al., 2019).

Of specific interest for this study are thermodynamic profiles from CDAAC's 1D-Var retrieval, described in Wee et al. (2022), since they are used as target data for training and evaluating our proposed DL retrieval framework.

## 3.2 Artificial Neural Network

The DL framework used in this study is comprised of an ANN in the form of a feed-forward multilayer perceptron (MLP).
ANNs are supervised neural networks, which map input data to a given output (label or target value) based on experience gained during the training on the data set (Gardner and Dorling, 1998). These models typically include three types of layers (input layer, one or multiple hidden layers, and an output layer) which are connected by its neurons. In a fully connected MLP, each neuron in a certain layer is connected to every neuron in the adjacent layer, whereas the strength of the particular connection is expressed by a numerical weight determined during training. The learning process is performed in an iterative
manner using some form of backpropagation algorithm. The input data are repeatedly fed into the neural network, multiplied by connection weights, summed up and passed to the next layer. Eventually, in the last layer, the model's error is estimated based on the differences between predictions and targets. In the next step, the calculated error is fed back and used to adjust the connection weights, minimizing the model error and producing output closer to the targets.

The number of hidden layers and neurons in these layers depend on the problem's complexity and on the amount of available
data. Along with other parameters, their optimal values are typically determined during a process called hyperparameter tuning. In atmospheric science, ANNs and various other types of ML models are used extensively nowadays, both as a framework for global weather prediction (Lam et al., 2022; Bi et al., 2022; Lang et al., 2024; Price et al., 2025) and for specific applications such as prediction of thunderstorms (Collins and Tissot, 2016) or tropical cyclone intensity (Jin et al., 2020). For further information, Chase et al. (2023) and Molina et al. (2023) provide good overviews of current applications in scientific studies
and operational forecasting.



**Table 1.** Overview of test hyperparameter combinations. The parameter values chosen for the final setup are marked bold.

| Parameter | Value |
| --- | --- |
| Number of hidden layer | **1**,2,3 |
| Number of neurons | **1000**,2000,2500, |
| Batch size | 100,250,500,**1000** |
| Number of epochs | 50,100,300,**500** |

## 3.3 Model setup

For this study, different setups have been tested during hyperparameter tuning, by varying parameters such as the number of hidden layers, number of neurons in these layers, as well as the batch size and the number of epochs for which the model is trained. An overview of the conducted test runs is given in Table 1. The final model setup, implemented using the PyTorch
library (Paszke et al., 2019), consists of input and output layers as well as one hidden layer with 1000 neurons. It was trained for 500 epochs using a batch size of 1000 and an adaptive learning rate. The Adam optimization algorithm (Kingma and Ba, 2014) was used to adjust weights and learning rates during the training process and the mean squared error (MSE) metric was applied as a loss function.

## 3.4 Feature setup

All feature setups tested during the development of the AROMA retrieval in the training and testing process are detailed in Table 2. These setups were compared in terms of performance on the test data set. These tests revealed approximately equal performance for different setups using either BA or N profiles individually as well as both of them together in combination with other independent features such as geographical coordinates or signal-to-noise ratio (SNR) indicators. For the final setup, we decided to employ the BA stand-alone feature setup. It makes use of ionosphere-corrected BA from the atmPrf and the
corresponding local spectral width (LSW) profile, impact height (IMPH) of the observations, as well as additional information extracted from the atmPrf file format provided by CDAAC. This setup maximizes the retrieval's independence from auxiliary data, since using BA itself avoids a BA optimization procedure commonly used in the retrieval of N which relies on high-altitude climatologies. LSW can be considered a proxy of BA uncertainty in the lower troposphere (Hocke et al., 1999; Gorbunov et al., 2006; Sokolovskiy et al., 2010). It has been used to estimate random errors (Sjoberg et al., 2023) and refrac-
tivity biases (Pham et al., 2024) of RO observations as well as to improve quality control procedures in NWP DA (Liu et al., 2018). IMPH represents the vertical coordinate associated with the BA and LSW profiles. The additional features utilized for the final setup are listed below.

     – Latitude and Longitude of the tangent point.

     – SNR: The SNR values at 80 km altitude on the L1/L2 carrier frequencies (snr1avg, snr2avg) as well as their lapse rates
between 40-80 km (snr1del, snr2del) are used.



**Table 2.** Different feature setups for the AROMA model investigated in this study.

| Setup | RO missions | Profile features | Scalar features |
|-------|-------------|------------------|-----------------|
| Final | C2, Spire | BA, LSW, IMPH | Lat, Lon, SNR, IRS, STDV |
| Test 1 | C2, Spire | BA, LSW, IMPH | Lat, Lon |
| Test 2 | C2, Spire | N, BA, LSW, IMPH | Lat, Lon, SNR, IRS, STDV |
| Test 3 | C2, Spire | N, IMPH, Lat, Lon | Lat, Lon, SNR, IRS, STDV |

- STDV: This parameter denotes the standard deviation of the oscillations of the difference between observational and standard climatology bending angles between 60-80 km. It was found to be a useful indicator for the BA retrieval uncertainty in the upper atmosphere (Sjoberg et al., 2023) and thus was added to the AROMA feature setup.

- IRS: This parameter represents a binary flag indicating rising or setting occultation (-1: rising; +1 setting).

## 3.5 Training and validation setup

As in any ML framework, the available data sets have to be split into training and test data. The training data set is utilized by the respective model (here an ANN/MLP) to learn underlying patterns and dependencies between input features and targets, by the assignment of weights between the connecting layers. The test data is separated and left out of the training process, in order to independently assess the model performance on unseen data afterwards.

For the AROMA retrieval, we make use of all data from C2 and Spire Global freely available from CDAAC during the period 01.01 - 26.10.2022 (300 days). This equates to approximately 1.5 million profiles, which are randomly split by a 80/20% ratio between training and test data, a typical ratio in ML studies. External validation is performed against ERA5 and radiosondes. PlanetiQ data are used as inputs for the external validation using ERA5 (Section 4.2.1) in order to investigate the performance of AROMA for observations from an RO mission not seen during training.

## 3.6 Data pre-processing

Pre-processing of input data is a common and vital step for most frameworks and applications using ML and DL models. For the AROMA retrieval, the first pre-processing step is an initial data screening, in order to exclude profiles of bad or questionable quality. This is done by checking the respective quality flags provided by CDAAC in the atmPrf and wetPf2 files, filtering out single profiles flagged as bad retrievals.

In a second step, vertical interpolation of input features, target variables (pressure, temperature and specific humidity), and reference data sets to a common altitude grid, defined in mean sea level height (MSLH), is carried out. Since the focus of this study is on the lower troposphere, the top height level is set to 20 km, while the lowest level is set to 0.5 km, using a grid step of 50 m. As not all RO profiles reach this lowest level, we disregard those that cut off at higher levels. Although other approaches are conceivable, this represents a compromise between avoiding extrapolation errors, while still ensuring a large sample size resolving the lower moist troposphere.




Furthermore, all input features are scaled and normalized using the MinMax scaling routine available from the scikit-learn library (Pedregosa et al., 2012). Each input feature at each vertical level is linearly scaled to a fixed range, where the largest occurring data point corresponds to the maximum value and the smallest one corresponds to the minimum value. This represents common practice in ML studies, which increases stability and speeds up convergence during the training process.

In addition, we also scale the target data for training the AROMA model. Although this step is often unnecessary and thus omitted for other ML setups, it is useful here, since the values of our target vectors are of very different magnitudes, from pressure at low levels ($\approx 1000$ hPa) to very small values of humidity at higher altitudes ($\approx 0$ g/kg).





## 4 Results

This section presents some initial results from the AROMA retrieval and their validation using both 1D-Var results from the
test data and external data sources such as ERA5 and RAOB soundings.

In the following, we show both overall error statistics as well as vertical error profiles, averaged over all respective valida-
tion samples. In the external validations, we directly compare AROMA with the standard 1D-Var profiles from CDAAC. All
these comparisons and respective error statistics are made both for absolute and relative differences. Relative differences are
calculated by normalizing the absolute differences with the average vertical profile of each retrieval parameter over all profile
samples of the entire test set.

### 4.1 Internal validation

As a first step, an internal validation of the AROMA is carried out. This represents a performance assessment on the test data
set, as it is common practice for ML studies.

During the training process, the parameter-specific RMSE values were monitored and the best performing model was chosen
for the evaluations below. The achieved metrics of this final setup, as described in Section 3.4, are presented in Table 3.
Furthermore, we show both absolute (top panel) and relative vertical error profiles (bottom panel) in terms of bias (solid lines)
and standard deviation (STD, dotted lines) for the internal validation in Figure 1. The metrics given in Table 3 reveal an almost
bias-free performance and very high correlation between vertical profiles for all three retrieval parameters of the AROMA
retrieval in comparison to CDAAC. In terms of absolute STD, and respectively RMSE, pressure retrieval errors are nearly 4
hPa at the lowest altitudes. However, in terms of relative differences, this only accounts for less than 0.5% error compared to
CDAAC's 1D-Var retrieval.

The temperature retrieval shows a slight negative bias compared to CDAAC and more variation in the vertical bias distribution,
which nevertheless always stays below 0.5 K. The STD profile shows a similar shape as for pressure in the lower troposphere
with STD values up to 2.5 K but also an area of larger deviations in the lower stratospheric region. Again, these represent small
relative errors which do not exceed 1% in any altitude region.

For specific humidity, AROMA shows a small negative bias in the lowest 2 km and the typical gradual increase in STD with
decreasing altitude. However, due to the relatively low magnitudes of humidity at higher altitudes, deviations between the
retrievals lead to larger relative errors of up to 40%.

Overall, CDAAC and AROMA retrievals agree very well for the majority of profile samples and altitude regions. For pressure
and temperature, relative errors are below 1% in the entire domain, whereas specific humidity depicts larger relative errors
which are growing with height. Additionally, the average correlation between single profiles of CDAAC and AROMA included
in the test data set is very high, with R values reaching up 0.99.



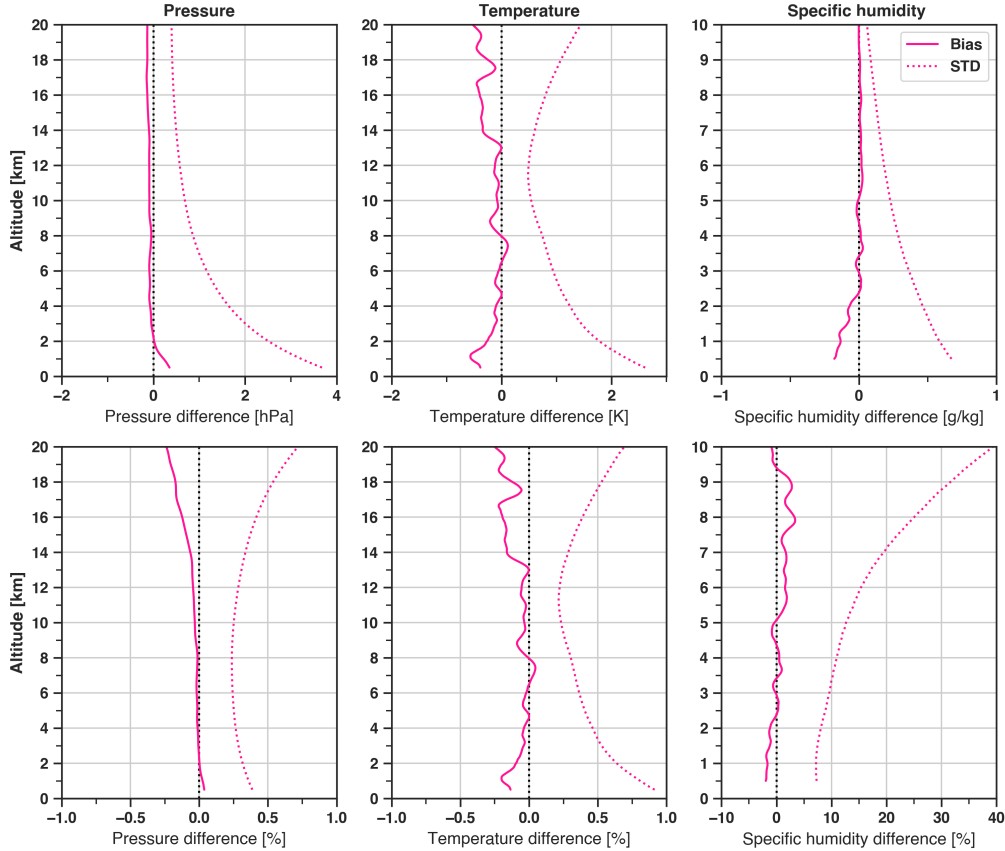

**Figure 1.** Vertical distribution of bias (solid lines) and standard deviation (dotted lines) values for deviations from wetPf2 for the test data set. Shown are absolute (top panel) and relative differences (bottom panel) in pressure (left) and temperature profiles (middle) up to 20 km MSLH, as well as specific humidity profiles (right panel) up to 10 km MSLH.

**Table 3.** Results of internal model validation on the test data set in terms of the absolute and relative statistical measures bias, STD, RMSE, and R.

|      | Pressure [hPa / %] | Temperature [K / %] | Specific humidity [g/kg / %] |
|------|--------------------|---------------------|------------------------------|
| Bias | -0.072 / -0.06     | -0.195 / -0.08      | -0.007 / -0.75               |
| STD  | 1.021 / 0.34       | 0.983 / 0.41        | 0.134 / 23.29                |
| RMSE | 1.291 / 0.37       | 1.119 / 0.46        | 0.221 / 26.82                |
| R    | 0.999              | 0.999               | 0.988                        |





**Table 4.** Results of external validation of C2 profiles using ERA5 in terms of RMSE for pressure, temperature, and specific humidity. The RMSE values are always shown for both retrievals, in the order CDAAC/AROMA.

| Profile | C2 | Spire | PlanetiQ |
|---|---|---|---|
| Pressure RMSE [hPa] | 1.05 / 1.41 | 0.92 / 1.81 | 0.74 / 1.61 |
| Temperature RMSE [K] | 1.03 / 1.31 | 0.78 / 1.45 | 0.81 / 1.67 |
| Specific humidity RMSE [g/kg] | 0.39 / 0.46 | 0.27 / 0.36 | 0.27 / 0.39 |
| Number of profiles | 38160 | 57422 | 11396 |

## 4.2 External validation

### 4.2.1 ERA5

In addition to the standard procedure of evaluating the model's performance on the test data set, two external validation experiments have been performed. The first experiment is a comparison of CDAAC and AROMA retrievals against ERA5 reanalysis products. Vertical profiles of pressure, temperature and specific humidity from 30 days (DOY 95-125) in 2023 were obtained from ERA5 using the interpolation procedure detailed in Section 3.6. This time period has been chosen due to the availability of data from all three missions analyzed in this experiment: C2, Spire and PlanetiQ.

Figures 2, 3 and 4 show vertical profiles of bias and STD of differences between ERA5 and CDAAC/AROMA retrievals. In general, the results indicate very similar overall performance of both retrievals, with slightly lower STD values for CDAAC throughout the entire domain. For pressure, these performance differences are most pronounced in the upper part of the domain (STD of 0.5-1% between 14-20 km), whereas for temperature, differences are largest in the lower troposphere (STD between 0.5-1.5% at 0.5 km). For specific humidity, biases stay at a rather constant value of about 2-3% throughout the entire domain. As already seen for the internal validation, relative STD errors in specific humidity reach large values (between 30-70 %) 280 compared to the other retrieval parameters.

The comparison of results for the different RO missions reveals somewhat similar performance levels of both retrievals for all missions. Vertical error profiles for C2 reveal a pressure bias of up to 1% at higher altitudes (above 12 km), which is present in both retrieval types. At these altitudes, biases exceed the retrieval STD for C2, whereas for the other missions the opposite can be observed. Retrievals of Spire profiles obtained from CDAAC are essentially bias-free for all three parameters, while 285 AROMA shows small bias variations at some altitude regions. AROMA STD values are slightly larger than those of CDAAC for all parameters, with differences of 0.5-1% for pressure and temperature and 5-10% for specific humidity.



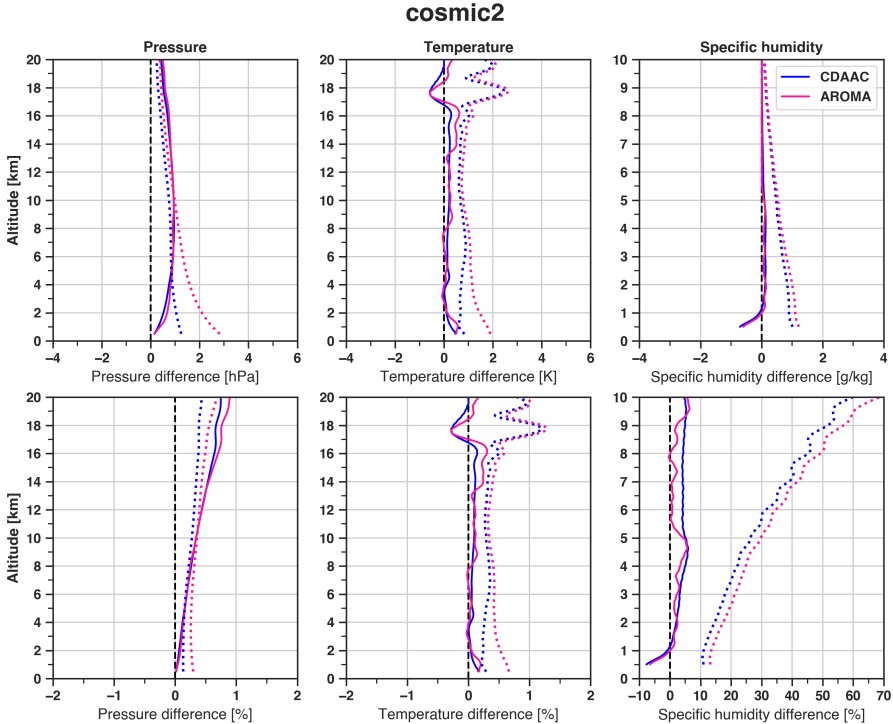

**Figure 2.** Vertical profiles of bias (solid lines) and standard deviation (dotted lines) values for differences from wetPf2 (blue) and AROMA (pink) to ERA5 for C2. Shown are pressure (left) and temperature profiles (middle) up to an altitude of 20 km, as well as specific humidity profiles (right panel) up to 10 km.

**Table 5.** Results of external validation of C2 profiles using ERA5 in terms of RMSE for pressure, temperature, and specific humidity. The RMSE values are always shown for both retrievals, in the order CDAAC/AROMA.

| Profile | C2 | Spire | PlanetiQ |
|---|---|---|---|
| Pressure RMSE [%] | 0.52 / 0.63 | 0.38 / 0.56 | 0.39 / 0.63 |
| Temperature RMSE [%] | 0.46 / 0.57 | 0.35 / 0.61 | 0.41 / 0.63 |
| Specific humidity RMSE [%] | 38.19 / 44.72 | 21.16 / 27.34 | 26.70 / 33.21 |
| Number of profiles | 38160 | 57422 | 11396 |

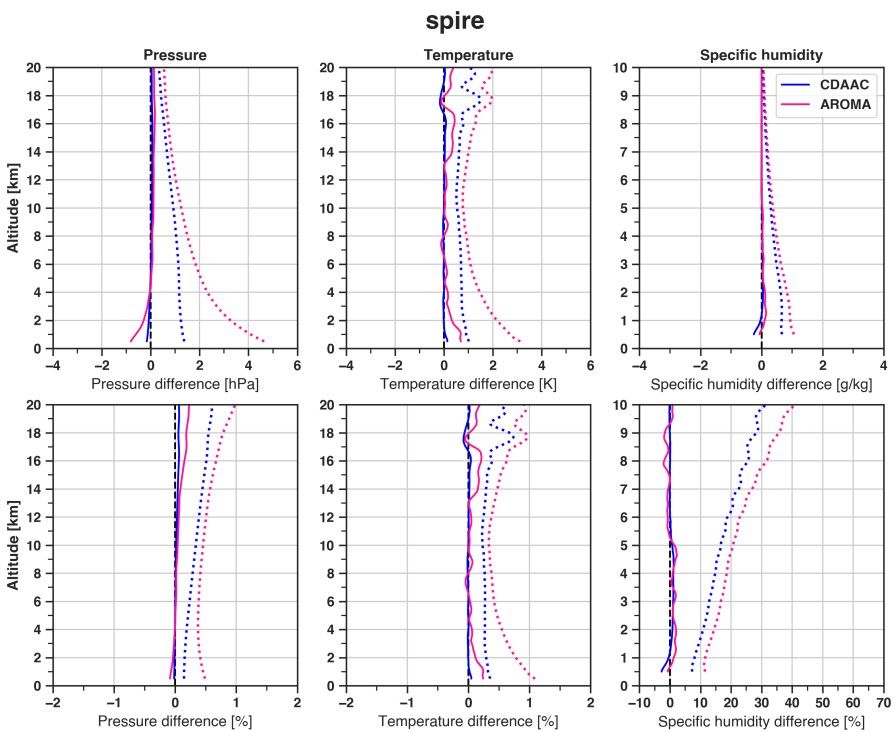

**Figure 3.** Same as Figure 2 but for Spire profiles.

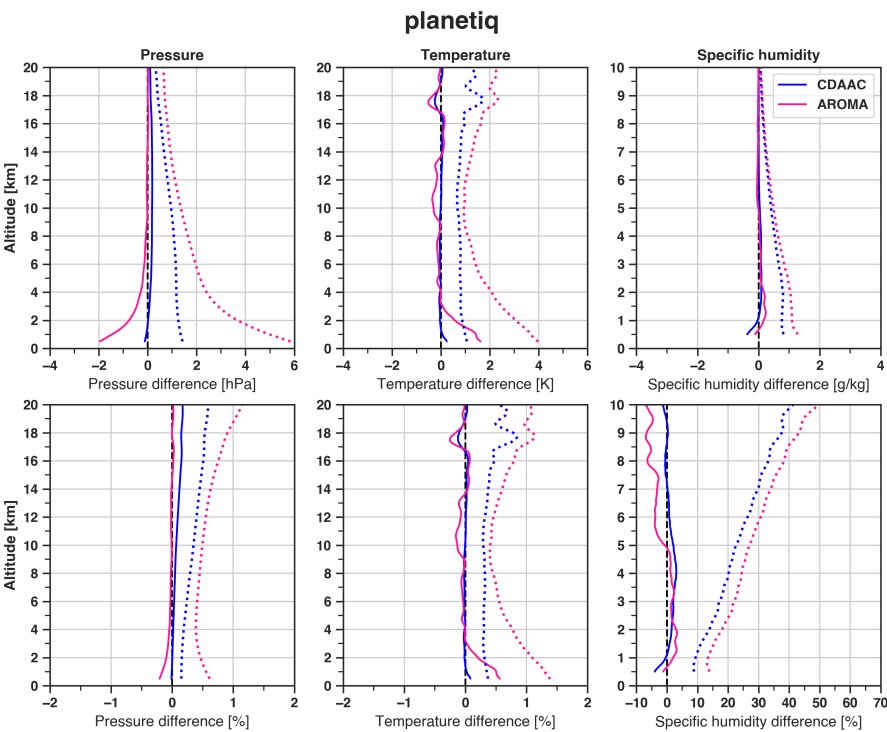

**Figure 4.** Same as Figure 2 but for PlanetIQ profiles.



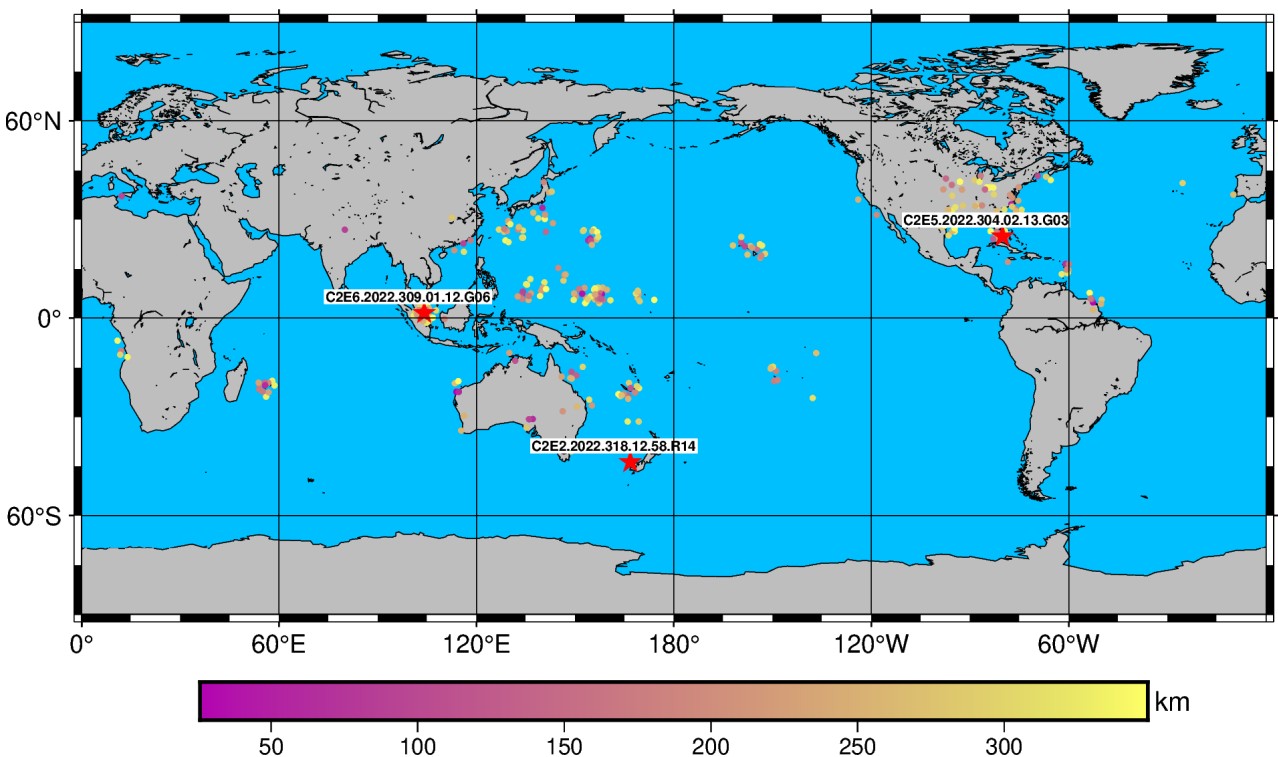

**Figure 5.** Locations of collocated RO-RAOB pairs utilized for external validation. The red stars mark the locations of the three specific profiles shown in Figure 7.

**Table 6.** Results of the external validation using 588 RAOB profiles in terms of RMSE for pressure, temperature, and specific humidity.

|  | CDAAC | AROMA |
| --- | --- | --- |
| Pressure RMSE [hPa / %] | 1.78 / 0.96 | 2.10 / 1.02 |
| Temperature RMSE [K / %] | 1.68 / 0.81 | 1.92 / 0.90 |
| Specific humidity RMSE [g/kg / %] | 1.07 / 102.06 | 1.09 / 103.09 |

### 4.2.2 Radiosondes

In addition to ERA5, we also validate the model performance using RAOB profiles. In total, we make use of 588 globally
distributed RAOB soundings collected between 27.10 - 14.11.2022 (DOY 300-335). These specific profiles are pre-processed
and collocated with corresponding C2 profiles using the criteria outlined in Section 2.3. Their exact locations are shown in
Figure 5.

Results of this RO-RAOB comparison are presented in Figure 6 and Table 6. They depict a very high agreement between
RAOB and RO profiles in general, with bias and STD values for pressure and temperature seldom exceeding 1%. This is also
reflected in the overall RMSE values given in Table 6, where even absolute differences in RMSE are less than 0.5 hPa / K for





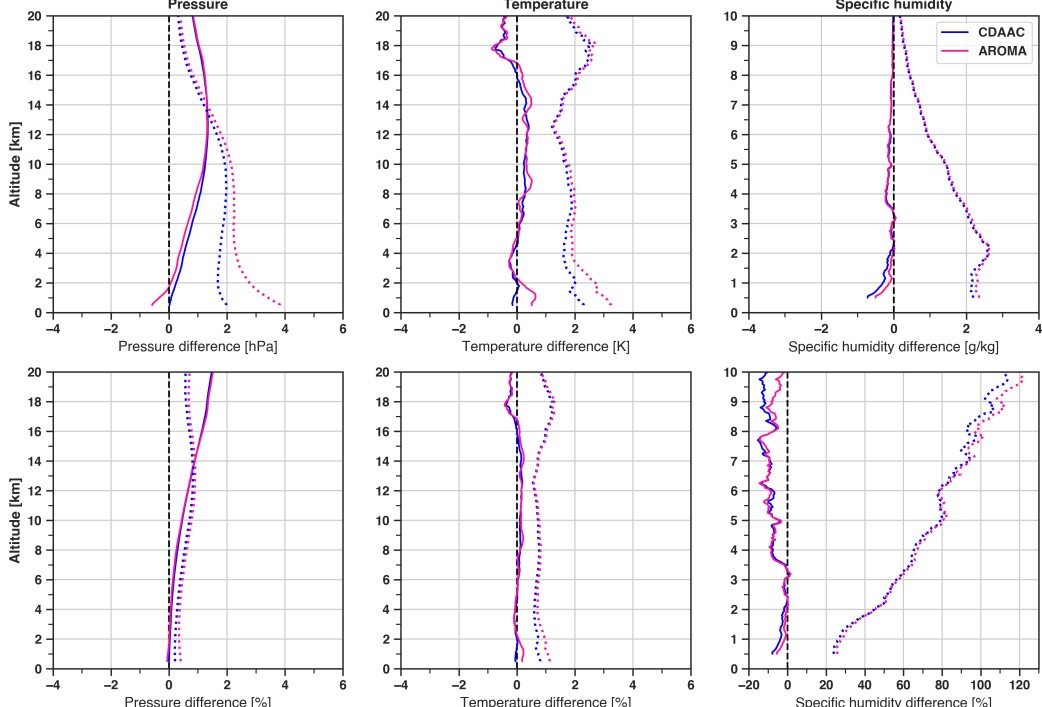

**Figure 6.** Vertical profiles of average RAOB-RO differences for pressure (left), temperature (middle), and specific humidity (right) retrievals from CDAAC (blue) and AROMA (pink).

the pressure and temperature retrievals and are as low as 0.02 g/kg for specific humidity. For specific humidity however, both retrieval methods show a distinct dry bias above 3.5 km and relative STD values again reach values in the upper part of the domain, as seen for previous validation experiments.

The retrievals from CDAAC and AROMA are in remarkable agreement with each other, with negligible differences in relative
errors for temperature and pressure in the lower troposphere as well as for specific humidity above 8 km, where AROMA shows slightly degraded performance. The dry bias mentioned above is slightly reduced for AROMA in this altitude region.

In addition to the overall RAOB validation, absolute differences for three specific collocation pairs are presented in Figure 7 relative to the respective RAOB soundings for the CDAAC and AROMA retrievals. Detailed information about location, collocation distance, and average metrics for these profiles are given in Table 7. They include two mid-latitude profiles (24.9°N
and 43.8°S latitude) as well as a profile observed in the tropics (1.6°N latitude).

The first profile, shown in the upper row of Figure 7, is located at 24.9°N, 80.1°W, just off the coast of Florida, USA. AROMA and CDAAC show good overall agreement with this radiosonde, although a distinct bias is visible in the pressure profiles.





**Table 7.** Detailed location information and vertically averaged RMSE for three specific C2 profiles collocated to RAOB soundings. The RMSE values are always shown for both retrievals, in the order CDAAC/AROMA.

| Profile | C2E2.2022.318.12.58.R14 | C2E6.2022.309.01.12.G06 | C2E5.2022.304.02.13.G03 |
|---|---|---|---|
| Latitude [°] | 24.8828 | 1.5610 | -43.7921 |
| Longitude [°] | -80.1001 | 104.1437 | 166.7684 |
| Distance [km] | 100.789 | 37.490 | 270.302 |
| Pressure RMSE [hPa] | 1.95 / 2.19 | 0.39 / 1.24 | 1.06 / 1.53 |
| Temperature RMSE [K] | 0.88 / 1.07 | 1.19 / 1.26 | 1.99 / 1.82 |
| Specific humidity RMSE [g/kg] | 0.33 / 0.29 | 0.28 / 0.21 | 0.30 / 0.37 |

Except for a small region between 4-7 km altitude, the pressure and temperature retrievals from CDAAC and AROMA are in almost perfect agreement with each other. For specific humidity this level of agreement is present for the whole profile.

The second profile (middle row of Figure 7) is located close to the east coast of Malaysia (1.6N, 104.1E). For this collocation pair, large deviations to the RAOB temperature profile exist in the lower stratosphere (17-20 km altitude). Apart for this region the agreement between RO and RAOB is generally high, especially for the CDAAC retrieval. AROMA shows some larger pressure discrepancies in the lower troposphere, but for temperature and specific humidity, the retrievals again show very high relative agreement.

The last profile (bottom row of Figure 7) was observed over the Tasmanian Sea, off the west coast of New Zealand (43.8S, 166.8E). Again some large differences in RAOB and RO temperature profiles are present, this time in lowest 2 km. CDAAC and AROMA retrievals again agree very well at higher altitudes, but from 8 km altitude downward, pressure and temperature retrievals start to deviate.

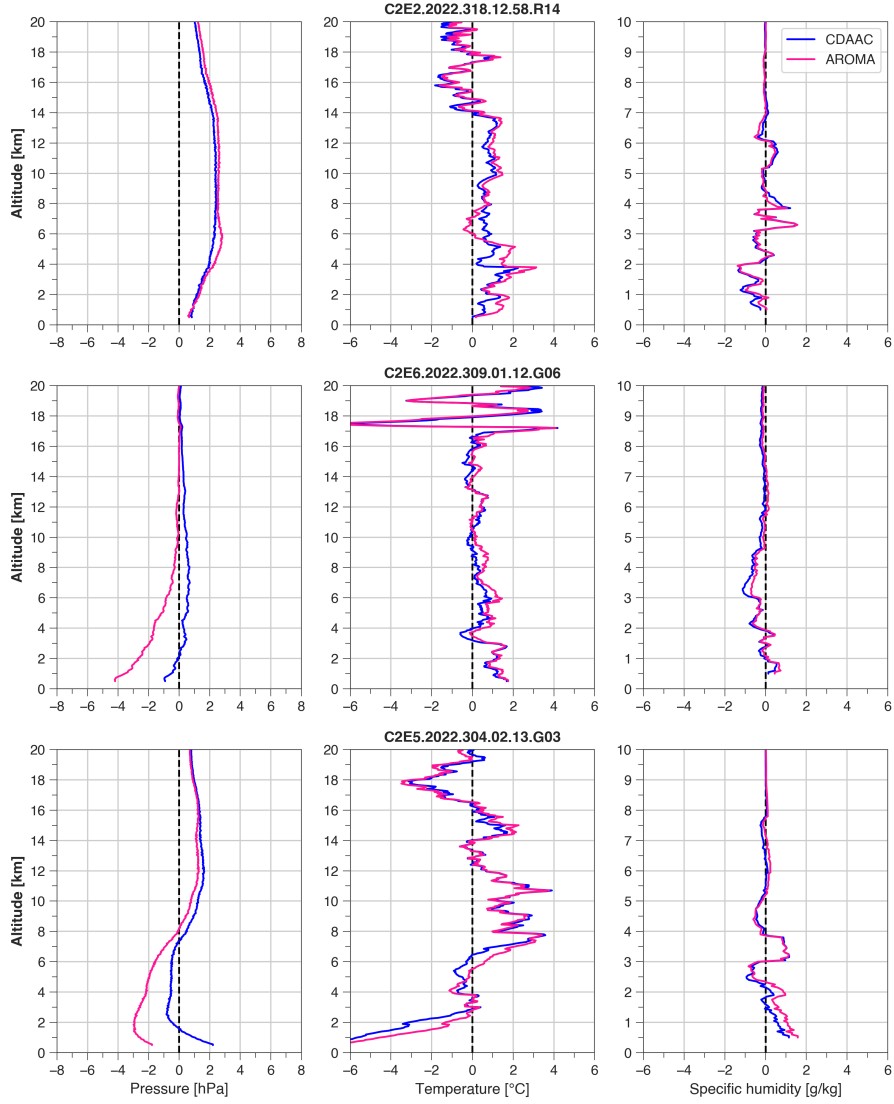

**Figure 7.** Vertical profiles of absolute differences in pressure (left), temperature (middle), and specific humidity (right) between RAOB and CDAAC (blue) and AROMA (pink) for the three specific RAOB-RO pairs detailed in Table 7.





## 5 Discussion

In the last sections, we have introduced a new DL-based RO retrieval method and presented some initial results, in comparison to the standard approach and external validation data.

In the first step of the validation procedure, the AROMA model was applied to the test data set. This test set consists of approximately 300000 randomly chosen RO profiles provided by CDAAC in the form of atmPrf and wetPf2 files. These profiles have been excluded from the training data in order to independently validate model performance. The average statistics shown in Table 3, calculated from differences between AROMA and CDAAC 1D-Var retrievals, indicate a good agreement between the two retireval methods. The retrievals of all three parameters are practically bias-free with respect to CDAAC and correlation coefficients all reach levels of 0.98-0.99. For pressure and temperature, STD values are highest in the lowest (0.5-4 km MSHL) and uppermost part (14-20 km MSHL) of the altitude domain. However, all these relative errors are small and do not exceed 1% in any altitude region. For specific humidity, absolute errors are small, but relative errors increase significantly with altitude as the total moisture decreases. These specific humidity errors may be due to instabilities in the retrieval. Furthermore, these error profiles indicate a slight decrease in agreement between the retrieval methods in the lower troposphere. This is common for RO observations, since the uncertainty of BA retrieval methods increases due to the increasing presence of moisture and its associated spatial gradients. Some of these discrepancies also represent the errors of imperfect training data, i.e. 1D-Var retrievals.

In the second validation step, CDACC and AROMA retrievals were compared to vertical profiles interpolated from ERA5 reanalysis data. This comparison was carried out separately for different RO missions (C2, Spire, and PlanetiQ), in order to analyze performance differences between missions included or not included in the training data set. Overall, both retrieval methods are in good agreement with ERA5 profiles and largely similar error statistics were obtained for all constellations. CDAAC slightly outperforms AROMA in most cases, a fact which is also visible in vertically averaged RMSE values given in Tables 4 and 5. However, errors of AROMA retrievals are generally small and on a comparable level to CDAAC. This holds true even for PlanetIQ data, which were not included in the training data set.

Some considerations regarding this comparison have to be kept in mind. Firstly, both C2 and Spire data are assimilated into the ERA5 reanalysis. Furthermore, ECMWF NWP background fields, assimilating many of the same observations, model parameter, etc. used in ERA5, are used in the CDAAC 1D-Var retrieval. Both these facts imply that any comparison between them cannot be considered fully independent. CDAAC retrievals are therefore more likely to show smaller discrepancies with respect to ERA5 than AROMA, which uses a very limited amount of NWP background knowledge (exactly the 300 days of training data) for the retrieval process. This may also be why, despite not being assimilated into ERA5 yet, the differences for PlanetiQ are of similar magnitude between AROMA and CDAAC retrievals.

In a final validation step, AROMA retrievals were compared to approximately 600 RAOB observations. Overall, this external validation reveals almost similar performance of the CDAAC and AROMA retrievals with respect to the RAOB soundings. The average RMSE for both retrieval methods is around 1% for pressure, and even slightly lower for temperature retrievals. Relative errors in specific humidity are again around 20% at lowest high levels and gradually increase to high values (>



100%) in the upper part of the domain, due to reasons already explained before for the internal validation. For the temperature
retrievals, slightly better statistical values are obtained by CDAAC, but overall differences to AROMA are as small as 0.25 K in
RMSE. Although the vertically averaged bias is approximately equal to 1D-Var results, the standard deviation, and respectively
RMSE, is higher (1.78 vs. 2.10 hPa). Furthermore, the AROMA performance has been analyzed for three specific RO-RAOB
profile pairs from different geographical regions (mid-latitudes and tropics). Like the overall RAOB validation results, they
show that AROMA results are largely comparable to those of CDDAC, with somewhat larger absolute pressure errors in the
lower troposphere. Overall, the RAOB validation provides encouraging results, showcasing that AROMA retrievals can provide
thermodynamic profiles with almost identical accuracy as CDAAC's 1D-Var retrieval.

In comparison to past studies, we obtained somewhat similar performance to the results presented in Lasota (2021) or Hooda
et al. (2023), in terms of error statistics. However, their results are not fully comparable to ours, since both of these (and
other) studies used reanalysis data such as ERA5 as targets. Therefore, larger differences to the operational CDAAC products
are excepted, whereas for this study a high level of agreement with CDAAC retrievals is the optimal result per definition.
Furthermore, our results are valid for all three major thermodynamic parameters and on a global scale, in contrast to Lasota
(2021) and especially Hooda et al. (2023), whose studies are restricted to certain regions and/or the C2 constellation and its
spatial resolution ($\pm$ 45° latitude). Our setup implies that the retrieval is also usable for Spire RO data and to a certain degree
also for constellations not included in the training data, as shown in the ERA5 validation for PlanetiQ (Figure 4). However,
including profiles from more RO providers is likely beneficial for the retrieval performance and will help to build a more
generalized retrieval method in the future.

## 6   Conclusions and outlook

In this study, we presented initial results from a DL-based RO retrieval framework entitled Advancing the GNSS-RO retrieval
of atmospheric profiles using MAchine-learning (AROMA). AROMA allows for the computation of thermodynamic profiles
in the lower atmosphere using RO-specific quantities in a deep artificial neural network (ANN). The AROMA model has been
trained on almost a full year of data from the C2 and Spire RO missions, using vertical profiles of BA, LSW, and IMPH, as
well as other RO parameters, as input features and CDAAC 1D-Var results as target values for supervised learning during the
training process.

After tuning the hyperparameter setup, the overall accuracy of the method was assessed on an independent test data set not
seen during training. Results reveal a good retrieval performance, with bias values close to zero and very high correlation
between average vertical profiles from AROMA and CDAAC's 1D-Var retrieval. In terms of relative errors, both pressure and
temperature retrievals show small deviations to CDAAC, with bias values below 0.25% and STD values between 0.25 - 1%.

In addition to the internal validation, two external validation experiments have been carried out. The first one was a cross-
comparison between AROMA and CDAAC retrieval results with profiles interpolated from ERA5 reanalysis products. Results
from both methods are in good agreement with ERA5, with evaluations revealing relative errors mostly below 1% for pressure
and temperature retrievals. The direct comparison of the CDAAC and AROMA retrievals reveals small average differences,



with CDAAC showing a slightly superior performance. When interpreting these deviations, two possible reasons for them have been identified, both concerning independence from ERA5. The first one is the fact that ERA5 already assimilates C2 and Spire data. The second one concerns the fact that CDAAC's 1D-Var retrieval uses ECMWF NWP fields as background data,
which are also utilized for the production of ERA5. Therefore, higher agreement between CDAAC results and ERA5 is to be excepted, while AROMA produces retrievals with very limited external information, i.e. NWP fields used to produce CDAAC retrievals utilized for training.

The second external validation used profiles from 588 RAOB soundings collected during a period of approximately one month in late 2022. This fully independent analysis confirms that overall, AROMA provides similar retrieval performance
to CDAAC's 1D-Var method, with normalized error differences between the two retrievals being below 0.5% respectively.

These represent very satisfactory results, since by default the setup presented here will not necessarily increase retrieval accuracy, but rather independence and, potentially, near real-time availability. Compared to the CDAAC 1D-Var, as well as similar approaches followed by other processing centers, the AROMA model has the major advantage of a significantly reducing the retrieval's dependency on external meteorological data. Apart from the prior information that was used to generate the wetPf2
retrievals that were utilized during the training process, it does not need any external data to produce thermodynamic profiles from the selected feature set.

In future studies, several pathways to further improve the current AROMA setup can be explored. Although we have shown that a DL-based retrieval method can mimic results produced by the standard retrieval algorithm without external data, a refinement of our approach needs to be implemented by future studies. Based on the initial results presented in this study, one
major point will be a detailed investigation on the long-term performance of the setup using multiple years of data and different RO missions. This will allow to analyze the ability of AROMA to generalize over data from various time ranges and RO missions. Furthermore, the applicability of the setup for profile retrievals in severe weather events such as tropical cyclones and atmospheric rivers needs to be studied. Concerning the model setup, a larger training data set and different types of setups for input features and targets can be explored. Especially regarding target data, a smart setup definition will be necessary in order
to increase accuracy and reliability of RO retrievals, in particular in the lower troposphere. This might require a combination of reanalysis or model data with actual RO products as targets for ML, while still managing the fine balance between performance and independence of the resulting retrieval method. In particular, data records as provided by ROMEX will be explored for training and testing ML-based RO retrieval frameworks in the future.

*Code and data availability.* All RO data used in this study are available from the UCAR COSMIC Program at https://doi.org/10.5065/
T353-C093 (UCAR COSMIC Program, 2019). ERA5 data are available from the Copernicus Climate Data Store at https://doi.org/10.24381/CDS.BD0915C6 (Copernicus Climate Change Service, 2018). RAOB observations are available from the NOAA National Centers for Environmental Information, dataset NCEI DSI 6327_02. The trained AROMA model and test data sets are available via https://polybox.ethz.ch/index.php/s/TysDKbn5KMYJyYM.



*Author contributions.* MAR developed the research idea, main software implementation and prepared the draft manuscript (including formal
analysis, visualization, and writing). JS provided the collocated RAOB data and parts of the software implementation. All authors contributed
to the discussion of results as well as reviewing and editing the manuscript.

*Competing interests.* The authors declare that they have no conflict of interest.



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
