# Peer review of "Retrieval of thermodynamic profiles in the lower troposphere from GNSS radio occultation using deep learning"

_EGUsphere, 2025_

## Author Comment (AC3)

**Response to Referee 2**

Review of

Retrieval of thermodynamic profiles in the lower troposphere from GNSS radio occultation using deep learning

by Aichinger-Rosenberger and Sjoberg

General comments

The manuscript presents an artificial neural network (ANN) approach for the retrieval of temperature, pressure, and specific humidity profiles in the neutral atmosphere, using a proposed framework called Advancing the GNSS-RO retrieval of atmospheric profiles using MAchine-learning (AROMA). Model training is based on a large dataset of profiles from COSMIC2, commercial data from Spire, and CDAAC 1D-Var products, which serve as target values for pressure, temperature, and specific humidity. Validation is carried out against 1D-Var profiles from CDAAC (set not used during training), as well as ERA5 reanalyses, radiosondes, and commercial data from PlanetiQ receivers. While the authors report generally small errors and high correlation values when comparing model outputs to validation datasets, the analysis of the results lacks depth and the comparison to previous studies remains unclear.

I believe a comparison against previous studies is crucial. Including RMSE results would facilitate comparison with works such as Lasota et al. (2021) (while acknowledging differences between the studies). Although some quantitative values are provided, I find the analysis of results somewhat vague. The manuscript frequently uses terms like "agree very well," "very high agreement," and "similar overall performance" without specifying the range of agreement or indicating whether the differences are statistically significant. An assessment of which specific regions may be performing better or worse could also be beneficial.

In addition, the authors state that a key advantage of their approach is its independence from external meteorological data. The model is trained using 1D-Var products as reference data, which are themselves based on ECMWF background fields, meaning that the training data is not completely independent. The authors comment on this in the Conclusions section but still claim the AROMA's main advantage is the independence of external meteorological data. I think this limitation should be more clearly acknowledged and articulated in the manuscript.

Overall, the paper is well written and well structured. I think it is encouraging to see incremental progress in the application of machine learning techniques to GNSS-RO data. The study is relevant and contributes to the field by expanding the training dataset in terms of size and time coverage, incorporating commercial GNSS-RO data for training and validation. However, I think the presentation and discussion of the results needs to be improved and the novelty better articulated. Therefore, I suggest major improvements are needed before publication.

Thank you very much for your review and feedback, as well as all the valuable comments and suggestions on how to improve the manuscript. We will try to extend and strengthen the performance analysis carried out, based on these suggestions.

One important aspect of our approach is its level of independence from external meteorological data, especially in comparison to the classic 1D-VAR technique. Here we want to clarify the misconceptions which have shown up in the reviews.

We never claimed that our approach is fully independent of external data, anywhere in the manuscript. This is obvious, since the model is trained using the operational 1D-VAR results. However, its level of independence is significantly higher than for 1D-VAR, as it does not need any auxiliary data to run the model once it is trained. The level of independence is determined by the amount of data used during model training. This tradeoff also explains why limiting the amount of training data is a reasonable approach, although much larger data sets could be utilized. We will try to clarify this further in the revised manuscript.

Regarding quantitative language used in the manuscript, we agree that the frequent use of subjective descriptors is not appropriate. We will amend relevant statements that we find during our revision.

In the following, we have answered specific comments one-by-one.

Specific comments

L10: "from both internal and external validation". Is this a common terminology for ML studies?

We will change the wording throughout the whole manuscript to e.g. *Performance evaluation on (test data / ERA5 /RAOB)*

L24-26: The study explores an alternative approach to retrieving thermodynamic profiles from RO; however, the importance and practical utility of these profiles are barely discussed.

We will add information on various use cases of thermodynamic profiles

L54: The term "internal and external validation" is introduced for the first time in the main body of the manuscript. Apologies if I am unfamiliar, but this doesn't seem to be a commonly used terminology. If authors choose to keep it, it would be helpful to briefly define it here for clarity.

We will change the wording throughout the whole manuscript to e.g. *Performance evaluation on (test data / ERA5 /RAOB),* as also suggested by other reviewers.

L62: How similar or different is your ANN compared to previous studies?

The ANN used in this study is quite similar to what was used by Lasota (2021) and Hooda et. al (2023). However, we use a different set of hyperparameters and features on which the model is trained. We will comment on this in the revised manuscript.

L67: I recommend including a brief outline of the paper at the end of the Introduction section. This would help guide the reader and clarify the flow of the manuscript, especially given the multi-step nature of the proposed methodology.

This outline will be added to the manuscript.

L115: RAOB is defined twice in the manuscript but is not used consistently throughout the document. Please revise.

This will be corrected in the revised manuscript.

L118: Could the authors clarify the rationale for choosing a 500 km collocation distance for RO observations in this study? This value appears notably larger than the 200 km distance that is typically used in the literature.

The main motivation was to have a larger sample size for the chosen period. We know this threshold is large and might introduce representativeness errors. However, these errors would identically impact both the 1D-Var and AROMA error statistics. The actual absolute errors are not the most important information communicated in these comparisons. The focus of our interpretation is more on the relative differences between 1D-Var and AROMA, which are small to non-existent. Identical error levels as 1D-Var is, by definition, the best possible result to be achieved with our target setup.

Still, the representativeness errors may be overwhelming the actual differences between 1D-Var and AROMA, so we plan to extend the amount of RAOB observations included in the validation by extending the validation period. Then we will be able to have an equal (or larger) sample size while reducing the collocation distance to the tyipcal 200-300 km.

L136: Please provide a citation for the constant terms.

We will add a citation in the revised manuscript.

L164: With regard to the ANN, can you explain why you chose a feed-forward multilayer perceptron (MLP) over more modern alternatives? Could more complex or structured architectures (e.g., CNNs, RNNs) be better suited?

The focus of this study was not to necessarily explore the performance of new, more complex deep learning architectures, but rather to investigate the impact of additional data, as provided by commercial RO missions. Therefore, we stuck to the simple ANN architecture which already showed some promising results in previous studies. Nevertheless, we are currently exploring more complex models for the same task, but these investigations are out of scope of this work.

L165: The statement "ANNs are supervised neural networks," is not correct. Please revise to reflect that ANNs can be used in both supervised and unsupervised learning contexts.

This statement will be revised.

L175: With regard to hyperparameter tuning, this seems to be slightly more explored in Section 3.3 on model setup. However, I think more context should be provided on how this is done and what is the practice in other studies.

L182-188: In general, there is a lack of justification for the chosen ANN architecture and the hyperparameter tuning process. What is the reasoning behind the parameter values presented in Table 1? Would it have been feasible to test larger batch sizes or a greater number of epochs? What limitations have you encountered? It would be helpful if the authors could comment on why this particular combination was the most successful, as this insight could be valuable for future work in this area. Additionally, providing figures or metrics to support these results would strengthen the manuscript.

Regarding the last two points: We will extend the number of combinations we test for hyperparameter tuning and the documentation of RMSE and correlation metrics for those setups. Therefore, a small grid search will be carried out. Extensive hyperparameter tuning is not doable in this study because of computational and time resources.

L194: signal-to-noise ratio (SNR) is defined in L74. Please use it accordingly.

We will correct this in the revised manuscript.

L227: Is there a reason why the top height level is 20 km? Climate and other studies using the retrieved thermodynamic RO profiles use data only up to this height?

The reason is that for this work, the focus is on the lower atmosphere. We plan to extend this up to 60 km altitude in future studies, as for the profiles typically provided by CDAAC in the wetPf2 format.

Section 4: As noted in my general comments, it would be helpful to use a more specific metrics range instead of vague terms. Providing actual ranges for the reported agreement would make the analysis clearer and more informative.

We will reformulate the relevant sections and provide the actual ranges for the agreement between AROMA and the respective validation data source.

L259: There is a negative bias above 12 km in both temperature and pressure in AROMA. Is this observed in other studies as well? Can the authors comment on what could be causing these biases? Are there specific regions contributing to them?

Thank you, this is an interesting point. We did not investigate these biases in detail during this study, but we'll try to do so in the revised manuscript.

Figure 1, 2, 3, 4, 6, and 7: All these figures could benefit from adding a letter identification. Also, I think these plots would benefit from adding the RMSE profiles in addition to or instead of the STD. Confidence intervals would also be helpful to see in these plots. What binning size is used?

Thank you for these comments.

We plan to add these letter identifications to the revised manuscript.

We also plan to add RMSE to figures and tables.

On significance testing, we don't agree that demonstrating whether the differences between the CDAAC and AROMA methods of moist atmospheric retrieval are significantly different (or not) adds much useful information to the reader. There are 1.5 million input profiles to these STDV values. Even by subsampling only 1% of all profiles in each dataset, the criterion for a 5% significant difference is a relative difference of ~1% between the two STDV values at any level. Given Figures 2-4, we are therefore likely to find that the STDV differences are significantly different everywhere. This does not mean that AROMA does poorly, only that we are confident that the standard deviations are different.

Vertical step size in these figures and analyses is 100 m.

Table 4: not referenced in the manuscript. Please revise.

Thanks for the hint, Tables 4 and 5 will be referenced.

L331: "instabilities in the retrieval." Can you clarify what this means?

Although we have not investigated this in detail, we suspect that it is observation noise learned by the model, in an altitude region where on average only small amounts of moisture are present. We will elaborate in the revised manuscript.

---

## Author Comment (AC4)

**Response to Reviewer 3**

The manuscript presents a DL-based RO retrieval method (AROMA) for retrieving RO atmospheric profiles. It offers an alternative to the current 1D-Var methods used in many RO data processing centers. This data-driven retrieval could be very useful given its computational efficiency and independence of NWP background data. However, there are several major flaws in the experimental design, validation methods, result interpretation and presentation. In particular, the metric chosen for many evaluations is not scientifically meaningful, and the resulting statements are therefore not valid.

Thank you very much for your review and the valuable comments and suggestions on how to improve the manuscript. We will try to extend and strengthen the performance analysis based on your suggestions.

In the following, we have answered your specific comments one-by-one.

Major comments:

Given that AROMA is a data-driven method, the amount of data used is very important. I do not understand why the dataset is limited to 300 days in 2022 and only two missions. What is the reason of not using data for a longer period or additional missions? This does not make much sense considering the data-driven topic and low computational cost.

Further, given the variations among missions in terms of data size, penetration, global distribution, etc., extending the training over a longer period would allow for a more interesting exploration of performance dependence on mission, latitude, SNR, and other factors.

The reason for this choice is the trade-off between the model's ability to learn/generalize (typically better with larger data amounts) and independence of external input data (in this case NWP data used for the 1D-VAR results which serve as targets). Therefore, the goal should be to choose the smallest amount of data, while still ensuring sufficient model performance. However, for the revised manuscript, we have gathered a larger training data set by incorporating other periods and one additional RO mission (GeoOptics) in the setup. We will also do some further analysis on the performance dependence on missions (some was already done, e.g. by testing on PlanetIQ data) and latitude. We agree that a deeper exploration of the model's performance is an interesting avenue for future work.

Regarding the computational costs, it has to be mentioned that the effort for the model training is definitely not low (which we also never stated in the manuscript), and will scale up further with larger data sets. One might speculate that once the model is trained, running it is probably faster than classical 1D-Var. However, we don't make this statement in the manuscript since we don't have sufficient knowledge on the computational effort of 1D-Var (e.g. per profile) and thus can not compare directly.

L76, L225–230, how many profiles are left after the QC (e.g., quality flag QC, 0.5 km penetration QC, etc.)?

In the extended setup we will use for the revised study, there are approximately 3 million profiles left after QC.

"over 50% of profiles reaching below 200 m above the Earth's surface." This is C2, how about the penetration of Spire? Because the study aims to retrieve GNSS-RO thermodynamic variables in the lower troposphere, why do the authors cut the retrieval to 500 m given the 1D-Var retrieval produces as much information as possible? The authors are expected to address this issue with larger training data set. Also, a figure could be helpful to answer these questions.

This is true for C2, but not for other missions. Thus, a very limited amount of data would be available from Spire and GeoOptics (added in the revised study setup). Furthermore, the MLP used in this study requires input and output vectors of constant length, which makes a regular height grid necessary. As we want to avoid extrapolation, we chose a cut-off height of 500m as a compromise between the number of usable profiles and resolving the lowermost troposphere.

However, we would like to address this issue in a future study using either more data (again an independence trade-off) or (better) with a more sophisticated model which allows for input vectors of varying length such as CNNs or transformers. For instance, we plan to investigate in a future study whether imputing missing data by filling it with zeroes enables AROMA to produce quality retrievals down to the surface.

"… provided approximately 8000–10000 profiles per day in January 2021". The number for the study period should be mentioned.

We will add this information for the study period.

The presentation could be improved, particularly in the introduction and description of the data, given that many different types are involved in this study. I found it difficult to follow. For example, the authors use CDAAC, wetPf2, and 1D-Var retrieval interchangeably. In Section 2.1, the authors should clearly introduce what CDAAC, atmPrf, wetPf2, 1D-Var, bending angle profiles, and thermodynamic profiles are, and then use these terms consistently in the following sections.

Thank you for this comment. We will introduce the datasets and fields we used and reduce the number of terms for the datasets we used to improve clarity.

Please have panel labels for all multi-panel figures.

We will add panel labels for all relevant figures.

Figures 1-4:

I understand that errors in bending angles are normalized by the observations and expressed as percentages, but reporting percent error in temperature or pressure is unusual and misleading. Think about the observation error specification of RO and radiosonde temperature in data assimilation. For example, the C2 STDV (Figure 2, top middle panel) at 2 km is about 0.7 K and 1.5 K for CDAAC and AROMA, respectively. Reporting AROMA's error as ~0.5% at 2 km (Figure 2, bottom middle panel) is therefore misleading. A more

appropriate approach is to calculate the change in STDV relative to a reference STDV, i.e., the STDV difference between the two methods normalized by CDAAC's method. In addition, the significance of such differences should be included in the revision.

This is a helpful critique. Our intention was to show the vertical variation of STDV more clearly by normalizing with respect to the mean of each field, in particular for specific humidity and pressure where the STDV values appear to asymptote to zero but are in fact relatively large. This was done for all three fields for consistency, but we agree that this subsequently led to misinterpretation of the findings.

The reviewer suggests taking the CDAAC method as our benchmark against which we should calculate differences of standard deviation of differences (STDV) from ERA5. The STDV of the CDAAC method is indeed one benchmark against which the performance of AROMA could be measured, but it's not clear what this suggested diagnostic represents beyond a percent change in the STDV relative to the CDAAC method. As the reviewer notes, the STDV values here do not necessarily represent observation error standard deviation (uncertainty) estimates, and thus this suggested diagnostic would not necessarily represent the relative change in uncertainty between the methods.

Nevertheless, comparison of relative differences is important. Because RMSE is the more typical metric for ML model evaluation, we opt to present relative change in RMSE between the CDAAC and AROMA methods. We will make this clear in the text that this is what is done. We will emphasize non-normalized differences as appropriate. And, we will remove or be cautious about the use of subjective modifiers about the metrics – "slightly," "very high," "small," etc. as the reviewer noted below –  throughout the paper.

On significance testing, we don't agree that demonstrating whether the differences between the CDAAC and AROMA methods of moist atmospheric retrieval are significantly different (or not) informs the reader about the performance of AROMA. There are 1.5 million input profiles to these STDV values. Even by subsampling only 1% of all profiles in each dataset, the criterion for a 5% significant difference is a relative difference of ~1% between the two STDV values at any level. Given Figures 2-4, we are therefore likely to find that the STDV differences are significantly different everywhere. This does not mean that AROMA does poorly, only that we are confident that the standard deviations are different.

Therefore, many statements are not valid, and I only list some of them,

L276, "In general, the results indicate very similar overall performance of both retrievals, with slightly lower STD values for CDAAC throughout the entire domain." The differences between CDAAC and AROMA are large. For example, the temperature STD difference for Spire is about 0.8 K and 1.2 K at 4 km and 2 km respectively.

L 293, "They depict a very high agreement between RAOB and RO profiles in general, with bias and STD values for pressure and temperature seldom exceeding 1%." Due to the inappropriate metric used, it is not a scientifically meaningful statement.

L382, "In terms of relative errors, both pressure and temperature retrievals show small deviations to CDAAC, with bias values below 0.25% and STD values between 0.25 - 1%."

We agree that the use of subjective descriptors is not appropriate for these statements. We will amend these statements, and others like them, that we find during our revision.

It is inconsistent that some figures show STDV values while the tables provide RMSE (e.g., Table 5 vs. Figs. 2-4).

We will make this more consistent throughout the manuscript, giving RMSE and STDV where appropriate.

The authors present figures for C2, Spire, and PlanetiQ, but provide no discussion of PlanetiQ. What is the purpose of Figure 4? I note that PlanetiQ's STDV is, on average, the largest among the three missions for AROMA. The authors should clearly describe these figures and provide a scientific explanation for these differences.

Figure 4 is intended to show the performance of AROMA against an RO mission that was not included in training and has notable differences in sampling, receiver, etc. from C2 and Spire. We will add discussion to Section 4.2.1 describing the contents of Figure 4 and additional discussion about our understanding of why the validation is different between missions.

Figure 5, the authors should explicitly describe what the colors represent in the caption.

Thank you, we will make this change.

L289, I do not understand why only limited RAOB data are used here. Why don't the authors use a larger dataset for verification? Again, a significance level is needed.

In similar studies such as Lasota (2021) and Hooda et. al (2023), an equal number of radiosondes were used for the verification. However, we plan to extend the amount of RAOB observations included in the validation by extending the validation period. Then we will also be able to have an equal (or larger) sample size while reducing the (now relatively large) collocation distance of 500km to 200-300 km.

We have addressed the need for significance testing in an earlier response.

Technical issues/Specific comments

L13: RO -> RO retrieval process.

We will correct that in the revised manuscript.

L21: Over the last decades?

L22 & many more: remove "()" in the reference citations. (e.g. Ruston et al. (2022)) -> (e.g. Ruston et al. 2022)

L25: remove the extra ")"

All the technical points above will be corrected in the revised manuscript.

L40: what exactly do the authors mean by "RO product"?

We have specified this now in the revised manuscript.

L56: water vapor pressure?

L75: "Large" is not proper for penetration.

L109: spatial -> horizontal; please also mention the vertical resolution.

L116,193,195, etc.: please use the acronyms (e.g., ROAB) if are defined already.

All the technical points above will be corrected in the revised manuscript.

L118–119, is there any justification to the 500 km/3h collocation threshold? Maybe some explanation or citation?

While 300 km, 3 hours is more common, this resulted in too few RAOB for meaningful results. We have expanded our RAOB comparison over a much longer period and thus will investigate whether we can lower the thresholds to the conventional values.

L132, Eq.1: all the parameters in the equation should be described.

We will define the terms.

L135, Eq.2: This classic equation should be given citation(s).

We will add *Smith and Weintraub (1953)* as a citation in the revised manuscript.

L149, Eq.3: xb is usually used to represent the background vectors and capital B (similar to O) usually represents the background error covariance matrix.

L150: it is "H" not H[x] that denotes a forward operator.

We will correct the manuscript following the above two comments.

L155: ROM SAF

We will correct that in the revised manuscript.

L165: it is not fully correct. Maybe something like "ANNs are used as supervised neural network in this study, which map input….."

We will correct that in the revised manuscript.

L176: it looks a bit weird to have the ANN and ML studies in the introduction section and then again here. These could be reorganized to improve the readability.

We will include this part directly in the introduction. This hopefully results in improved readability.

L221–224, It does not seem like a pre-processing. The authors only used profiles with "good" quality flags.  There was not any processing involved.

We will change the wording here. However, we now adopted a "real" pre-processing routine which uses e.g. median absolute deviation (MAD) filtering of input features and targets.

L226, the coordinate of the features is IMPH, how is it related to MSLH?

Impact height (IMPH) is the height of the closest point of the ray path to the Earth's center, whereas mean sea level height (MSLH) is the height of the signal's path above the mean sea level. Formally, they are related through the impact parameter $a$ (see Eq. 1) where the radius $r$ is (MSLH + $r_C$ + $r_{geo}$) and impact height is ($a$ - $r_C$ + $r_{geo}$). We have added wording to briefly describe this relationship.

L251–252: This sentence feels weird. Maybe change to "Figure 1 shows …."

We have changed that in the revised manuscript.

L 271: I do not think it is an "experiment" here.

We will rephrase that in the revised manuscript.

L 324, I do not see the use of atmPrf in any evaluation.

We will remove "atmPrf" in the revised manuscript.

L 324, 300000->300,000

This will be corrected.